# Insurer green finance under regulatory cap-and-trade mechanism associated with green/polluting production during a war

Fu-Wei Huang [1], Jyh-Jiuan Lin [2]*

1 Department of Banking and Finance, CTBC Business School, Tainan, Taiwan, 2 Department of Statistics, Tamkang University, New Taipei City, Taiwan

* 117604@gms.tku.edu.tw

## Abstract

The cap-and-trade mechanism affects firms' production and operation decisions and carbon emissions, making them move towards environmental sustainability. This article develops a contingent claims model to examine the impact of the regulatory cap-and-trade mechanism on the green finance strategy of an insurer during a war. Participating in the cap-and-trade scheme of the insurer that funds the borrowing firms also implicitly affects firm production and carbon emissions. The results show that increasing the green loans decreases the interest margin of the insurer but helps policyholder protection. The insurer is reluctant to provide green loans for the green borrowing firm and thus retards sustainable development. A stringent regulatory cap of the cap-and-trade mechanism raises the insurer's interest margin but hurts policyholder protection. From the perspective of the insurer's profit, regulatory cap efficiently derives insurer lending toward sustainability through borrowing-firm cleaner production. An increased war impacting the polluting borrowing firm increases the insurer's interest margin but harms policyholder protection, affecting insurance stability adversely. This research enriches related literature and knowledge concerning insurer green finance practices indirectly associated with cleaner production. The research also highlights the significance of the regulatory cap-and-trade mechanism that reflects cleaner production in affecting insurer performance during a war.

## 1. Introduction

One of the main concerns of climate change is carbon emission reduction, which is associated with the development of sustainability [1, 2]. The cap-and-trade mechanism stimulates cleaner production toward environmental sustainability [3]. However, sustainable development heavily relies on green finance, contributing to sustainability through borrowing-firm cleaner production. In the study, we argue that the most prominent insurers worldwide significantly prioritize environmental, social, and governance (ESG) issues in their investment practices. Life insurers demand more ESG transparency from borrowing firms and encourage them to adopt strategies that support the net-zero carbon targets in the Paris Agreement. Insurers are concerned about green lending with borrowing-firm cap-and-trade transactions in their

**Data Availability Statement:** All relevant data are within the paper and its Supporting Information files.

**Funding:** The authors received no specific funding for this work.

**Competing interests:** The authors have declared that no competing interests exist.

investment practice. This practice is expected to support the transition to a more sustainable economy. Besides, life insurers can use reinsurance for policyholder protection to move liabilities from ceding insurers to less regulated affiliated insurers. The operating insurers cede 25% of every dollar insured to their affiliates in 2012, up from 2% in 2002 [4]. Following the 2022 Russian-Ukraine war, the economy provides an opportunity to reinvestigate life insurance activities. Accordingly, A contingent claim model evaluating a life insurer's equity value and insurance stability is the article's aim. Chiaramonte et al. [5] argue that the literature about sustainability practices and insurance stability is rare. Our focuses are on a nexus of the explicit treatment of borrowing-firm credit risks, borrowing firms involved in the cap-and-trade mechanism, green lending, and affiliated reinsurance during a war on which the previous literature remains silent.

Our issue is essential. First, insurer green finance provides funds for borrowing-firm investments resulting in increased environmental benefits. Literature has investigated the effect of green credit policy on borrowing-firm performance [6]. However, only limited research examines the influence of profitability and policyholder protection brought by the insurer's green credit policy. Profitability is central to the insurers' strategic and the regulators' decisions concerning insurance stability. Policyholder protection is essential for most insurers; insurers often take it as a basis when assessing their performance relative to each other. For regulators contemplating insurance prudential regulation, knowing how insurer green loans stimulate cleaner production toward sustainability affects insurer profitability and policyholder protection during a war is paramount. Second, the cap-and-trade mechanism effectively reduces carbon emissions through cleaner production [7]. Considering the carbon-emission allowance transactions from the insurer funding viewpoint is interesting. Explicit treating borrowing firms' credit risk becomes crucial, including an allowance transaction regulation on which further limited studies focus. This aspect goes further for an insurer's green financing, contributing to ESG. Third, one of the primary motives of reinsurance is risk transfer. Affiliated reinsurers could reduce the marginal cost of issuing life insurance policies, improving retail market efficiency [4]. Risk transfer of risk sharing, particularly in imperfectively competitive insurance markets, is a critical strategy in the asset-liability matching management of an insurer. Notably, most insurers eagerly transfer risks to affiliated reinsurers during a war. Therefore, our study is a step forward in ESG: the nexus of borrowing-firm cap-and-trade mechanism toward sustainable development, capped green credit risk from borrowing firms, and affiliate reinsurance during a war.

Neumann and Shenhav [8] argue that the life insurance industry and insurance research are well developed but had not involved in active wars. Their argument also indicates that the war rapidly causes inflation but does not adversely affect the total profitability of the insurance industry. Our research also focuses on the life insurance issue during a war. We contribute to the literature by considering the following features. A life insurance policy market faced by the ceding insurer is imperfective competitive [9]. The optimal guaranteed rate determination for profit maximization becomes a crucial insurer rate-setting behavior. Explicitly considering the credit risk from borrowing firms and policyholder protection encourages the insurer to cede insurance to its affiliated reinsurer. Moreover, the insurer's green loans associated with the cap-and-trade mechanism support the transition to a more sustainable economy. Thus, our research is beyond the current literature by considering the nexus of the above issues.

The main questions we address are:

i. What are the most likely influences on the optimal guaranteed rate (and thus on the optimal insurer interest margin): green loans, regulatory cap of cap-and-trade mechanism stimulating cleaner production, affiliated reinsurance, or the war?

ii. How do the four factors affect policyholder protection (and thus insurance stability)?

The answers contribute to the broader literature on relationships among green lending, affiliate, and environmental performance by a contingent claim model with the insurer's guaranteed rate-setting behavior instead of the frequently used mode for insurer rate-taking. The model includes more realistic market conditions and a more appropriate behavioral model of large-scale insurers' rate setting.

The contingent claims approach to the corporate securities valuation treats a firm's equity as a call option on the firm's assets [10] and a short put option on the firm's value [11]. Brockman and Turtle [12] further formulate the corporate security valuation based on a path-dependent barrier option model, focusing on a premature structure. Briys and de Varenne [13] use a contingent claim approach to evaluate a life insurance company's equity. Grosen and Jørgensen [14] apply Briys and de Varenne [13] by introducing a premature state of an insurer's equity valuation. This article develops the model in the spirit of Briys and de Varenne [13], Brockman and Turtle [12], and Grosen and Jørgensen [14].

We develop a contingent claim model for evaluating the market value of the insurer's equity to answer the questions above. On the revenue side, the asset portfolio consists of low-carbon-emission lending (green loans) and high-carbon-emission (brown loans). The borrowing firms also participate in the cap-and-trade scheme encouraging cleaner production for carbon emission reductions during a war. The insurer then explicitly considers the credit risks from the high- and low-emitters as a capped one. On the cost side, the ceding insurer determines the optimal guaranteed rate of the life insurance policy for the insurance holding company. The ceding insurer also moves liabilities to its affiliated reinsurer, particularly for policyholder protection during a war. Thus, our model deals with a strategic asset-liability matching management for an insurance holding company, which is a step forward on sustainability and reinsurance in the literature.

There are several interesting results found in the research. First, increasing green loans toward borrowing-firm cleaner production under the cap-and-trade mechanism decreases the insurer's interest margin. Still, these increased green loans help policyholder protection, mainly when a stringent regulatory cap is. Green-loan engagement costs the insurer's profitability. Second, the rigorous regulatory cap of the cap-and-trade mechanism raises the insurer's interest margin, harming policyholder protection. The fund provider (the insurer) prefers the government to conduct a stringent cap policy, yielding the insurer's profits. In this case, the insurer is willing to get involved in green lending, making the borrowing firms improve their cleaner production for sustainable development. Third, the affiliated reinsurance costs the insurer by a reduced margin but benefits policyholder protection. Finally, increasing the war impact on the polluting borrowing firm makes the insurer more conservative, decreasing the insurance businesses at an increased interest margin. But this impact hurts policyholder protection and adversely affects insurance stability.

Our four findings above demonstrate the influences of green loans, the regulatory cap of the cap-and-trade mechanism, affiliated reinsurance, and a war on the insurer's performance in a borrowing-firm-insurer situation. Chen et al. [15] explore the impacts of the cap-and-trade transactions and financial gray rhino threats on insurer green performance. The research examines the effects of the government green subsidy with green trading on the insurer's lending decisions [16]. Huang et al. [17] analyze free riding and insurer carbon-linked investment. Li et al. [18] investigate the green loan subsidy, regulatory cap, and green technology of borrowing-firm environmental impacts on insurer green finance assessment. Accordingly, our findings contribute to the literature on insurer green finance by extensively exploring the insurer's green funding for the green and polluting borrowing firms participating in the cap-

and-trade mechanism during a war. The regulatory cap encourages borrowing firms to get involved in cleaner production operations toward sustainability. Therefore, the insurer's green finance might improve sustainable development depending on the cap-and-trade mechanism's regulatory cap.

## 2. Literature review and background

A substantial literature has emerged on environmental sustainability and financial reinsurance impacts insurer profitability and stability during a war. This section first discusses the green credit policy issue. Consequently, this section discusses the cap-and-trade mechanism issue. Finally, we discuss affiliated reinsurance. Considering the similarity between papers that focus on green lending, cap-and-trade transactions, and affiliated reinsurance, this section discusses the link between the three issues during a war based on a contingent claim model developed in this article.

Carbon emission transactions in the cap-and-trade scheme are a cost-oriented mechanism for carbon emission reductions. The government determines a cap to incentivize firms to reduce carbon emissions. Firms are carbon-allowance buyers required by the government that their carbon allowance emissions exceed the regulatory cap. The carbon emission buying for production permission costs the high emitters, which creates an incentive to get involved in reducing carbon emissions in production. The carbon emission selling benefits the low emitters since they have put efforts into reducing carbon emissions. Carbon buyers and sellers in cap-and-trade transactions may improve environmental development by reducing carbon emissions.

Previous studies, such as Nandy and Lodh [19] and Cui et al. [20], find that green credit policy helps firms' green transformation and sustainable development and banks avoid environmental risk. Zhou et al. [21] discuss the effect of corporate social responsibility (CSR) on financial performance from the green credit risk policy viewpoint. The study finds a deteriorated impact in the short term but a beneficial impact in the long run. However, few researchers explore the effects of a green credit policy on an insurance holding company, considering that borrowing firms participate in carbon allowance transactions. Therefore, different from the previous literature, this study aims to study the determinants of the life insurance policy's optimal guaranteed rate setting and policyholder protection, considering credit risk from high-/low-carbon emission borrowing firms. The research provides suggestions for understanding the role of the cap-and-trade mechanism and insurance stability, particularly during a war.

Some scholars have conducted product pricing and carbon emissions under carbon emission constraints. In an efficient cap-and-trade mechanism, the high emitter needs to buy carbon quotas for production permits while the low emitter sells the surplus allowances to earn profits [22]. Carl and Fedor [23] point out that 70% of cap-and-trade revenues are earmarked for green spending. Zhang et al. [24] explore how the cap-and-trade mechanism and customer's environmental awareness affect productive-firm carbon emission reduction and pricing strategies. Yang et al. [25] explore the cap-and-trade mechanism from the aspect of optimal product pricing decisions and green technology investment, especially investigating the effect of allowance allocation rules on product pricing, technology choice, and total emissions. Overall, the previous studies focus on production-firm performance under cap-and-trade mechanisms. While our paper discusses a cap-and-trade procedure, we focus on the insurer performance. Accordingly, we can elaborate on the lending-borrowing issue with cap-and-trade transactions, which provide references for investors and regulators.

The purchase of reinsurance is relevant to solvency and earnings, influencing the firm's economic value. For example, Koijen and Yogo [4] develop a ceding insurer-affiliated reinsurer model. Their model concludes that shadow insurance (i.e., affiliated reinsurance) could improve retail market efficiency by relaxing capital requirements. Chen et al. [26] investigate the multiple shadow insurance and policyholder protection relationship by developing a contingent claims model and finds that multiple shadow insurance enhances superior return performance and helps policyholder protection. However, the previous papers remain silent on asset management when considering the insurer's equity determination. Our study develops an asset-liability matching management model, considering credit risk from borrowing firms involved in environmental cap-and-trade transactions and affiliated reinsurance. This insight is crucial for asset-liability matching management of insurers. The management structure consists of external and internal control to evaluate the insurer's equity value, taking our analysis differently.

## 3. Basic setup

### 3.1 Conceptual framework

The article applies a situation consisting of two borrowing firms and one insurance holding company to a contingent claim model. The two borrowing firms include a high-carbon polluting firm (i.e., the polluting borrowing firm) and a low-carbon polluting firm (i.e., the green borrowing firm). Both firms participate in the cap-and-trade system with the cap regulation. The polluting borrowing firm for production permits must buy carbon emission allowances in the carbon trading market based on the regulatory cap. The low-carbon emitter can sell the surplus allowances also based on the regulatory cap to earn revenues due to its greenness. The regulatory cap is usually shrinking for carbon emission reductions for environmental improvement worldwide.

Regarding insurer green finance, we assume the insurance holding company consists of a ceding insurer (i.e., the insurer) and affiliated reinsurer (i.e., the reinsurer). For simplicity, the borrowing firms' funds are only from the insurer without considering external finance. The framework assumes borrowing firms and insurers make all the decisions in a single-period horizon during a war to enrich our modeling.

### 3.2 Theoretical model

At the beginning of the period, the borrowing firms and the insurance holding company have the following four balance sheets:

$$\text{polluting borrowing firm}: \ A_h = (1 - \theta)A + K_h \tag{1}$$

$$\text{green borrowing firm}: \ A_l = \theta A + K_l \tag{2}$$

$$\text{ceding insurer}: \ (1 - \theta)A + \theta A + \beta L = L + K_g$$

$$= \alpha[\theta A + (1 - \theta)A + \beta L] + (1 - \alpha)[\theta A + (1 - \theta)A + \beta L] \tag{3}$$

$$\text{affiliated reinsurer}: \ A_a = \beta L + (1 - \kappa)K_a = \alpha_a A_a + (1 - \alpha_a)A_a \tag{4}$$

where

i. $A_h$ = the polluting borrowing-firm investment, $(1-\theta)A$ = funded by the insurer's conventional loans and $K_h$ = its capital where $0<\theta<1$ = the insurer's asset portfolio distribution coefficient

ii. $A_l$ = the green borrowing-firm investment, $\theta A$ = financed by the insurer's green-linked loans and $K_l$ = its capital

iii. $L = \alpha[\theta A + (1-\theta)A + \beta L]$ = life insurance policies where $\beta L$ = the investment funded by proportional reinsurance with the condition $(0<\beta<1)$, $K_g = (1-\alpha)[\theta A + (1-\theta)A + \beta L]$ = capital, and $\alpha$ = leverage ratio (i.e., $(1-\alpha)$ = capital ratio)

iv. $A_a$ = the affiliate's investment, $\beta L = \alpha_a A_a$ = liabilities (i.e., reinsurance from the insurer), where $\alpha_a$ = the affiliate's leverage ratio (i.e., $(1-\alpha_a)$ = capital ratio), $(1-\kappa)K_a = (1-\alpha_a)A_a$ = capital, and $\kappa$ = capturing the initial capital level of the affiliate.

Eqs (1) and (2) demonstrate that the borrowing firms finance their assets with insurer loans and capital for environmental improvement purposes. Eq (3) explains that the ceding insurer's investment portfolio includes green-linked loans for the green borrowing firm, conventional loans for the polluting borrowing firm, and reinsurance investment toward sustainability. Eq (3) reflects the relationships between the borrowing firms and the ceding insurer. The reflection indicates how the insurer finances borrowing firms for environmental improvement, which is an early step forward in sustainability compared with the previous literature [3; Yang et al., 2020]. The ceding insurer funds the portfolio through life insurance policies and its capital. The insurer also invests reinsurance to its affiliate, considering unexpected shocks for stop-loss insurance such as a war. The investment strategy of the insurer is liability-driven, considering a war impact. In the affiliated reinsurer's balance sheet of Eq (4), the right-hand side $(\alpha_a A_a + (1-\alpha_a)A_a)$ is the total investments funded by reinsurance and the affiliate's equity capital $(1-\kappa)K_a$ (i.e., $0<\kappa<1$ considering capital regulation purposes). The affiliate's asset portfolio also consists of green-linked and conventional loans. For simplicity, the portfolio are not used for funding the borrowing firms in Eqs (1) and (2). We do not explicitly consider those two assets for the cap-and-trade transactions in our model. If the mathematical model would be slightly different from the one presented in this section, the conclusions resulting from the complexity would hold completely. The condition $\alpha>\kappa$ indicates that the affiliate can operate with less capital, not subject to risk-based capital regulation [4]. Thus, Eqs (1)–(4) allow investigating the relationships between cap-and-trade mechanism and reinsurance during a war considering insurer green finance toward sustainability.

To explicitly capture the borrowing-firm credit risk when evaluating the insurer's equity, we assume that the market values of firms assets are variant continuously over the period according to the so-called structural stochastic processes:

Polluting borrowing firm:

$$\frac{dV_h}{V_h} = [\mu_h - (b_h + \frac{b_h^2}{2})]dt + [\sigma_h + (b_h + \frac{b_h^2}{2})]dW_h \qquad (5)$$

Green borrowing firm:

$$\frac{dV_l}{V_l} = [\mu_l - (b_l + \frac{b_l^2}{2})]dt + [\sigma_l + (b_l + \frac{b_l^2}{2})]dW_l \qquad (6)$$

where

$V_h = (1+R_h)A_h$ = polluting borrowing-firm investment return where $R_h$ = return rate
$\mu_h$ = an instantaneous expected return rate on $V_h$

$\sigma_h$ = an instantaneous standard deviation of $V_h$

$(b_h + b_h^2/2)$ = structural breaks in return and volatility, capturing the war seriousness

$W_h$ = a Wiener process

$V_l = (1+R_l)A_l$ =green borrowing-firm investment return where $R_l$ = return rate

$\mu_l$ = an instantaneous expected return rate on $V_l$

$\sigma_l$ = an instantaneous standard deviation of $V_l$

$(b_l + b_l^2/2)$ = return and volatility structural breaks, capturing the war seriousness

$W_l$ = a Wiener process

Both equations emphasize the war impacts on borrowing-firm investments captured by structural breaks in reduced return and incremental volatility. We assume that a quadratic form of the structural break reflects the war's seriousness.

The cap-and-trade mechanism can encourage production firms (the borrowing firms in our model) to reduce carbon emissions [27] from their cleaner production. We can express the investment returns of borrowing firms involved in the cap-and-trade transactions as follows:

$$\text{Polluting borrowing firm}: \; M_h = V_h - (c_h - c_{cap})A_h \tag{7}$$

$$\text{Green borrowing firm}: \; M_l = V_l + (c_{cap} - c_l)A_l \tag{8}$$

where

$c_{cap}$ = rate of the cap-and-trade mechanism's regulatory cap

$c_h$ = marginal rate exceeding the regulatory cap and buying required carbon emission allowance $(c_h - c_{cap})\,A_h$, costing the polluting borrowing firm

$c_l$ = marginal rate below the regulatory cap and selling the carbon emission allowance $(c_{cap} - c_l)\,A_l$, benefiting the green borrowing firm

We now integrate Dermine and Lajeri [28] with Briys and de Varenne [13] into the model. The integrated model reveals the advantage of the explicit treatment of Eqs (7) and (8) when evaluating the insurer.

Literature on sustainability-linked responsibility may create higher profits [29] that the paper considers green finance for borrowing-firm cleaner production toward sustainability. Liu [30] also argues that green loans (loans for the green borrowing firm in our model) are increasing and safer than conventional ones (loans for the polluting firm). Moreover, we also introduce the guaranteed rate-setting behavior and affiliated reinsurance to complement the integrated model. The life insurance policy market is perfectly competitive such that the insurer is a guaranteed rate-taker [13]. This assumption does not apply to concentrated life insurance markets where insurers set rates, according to Hong and Seog [9]. Koijen and Yogo [4] also point out that with significantly gowning, since 2007, affiliated reinsurance has exceeded unaffiliated reinsurance.

Under the circumstances, the asset portfolio of the ceding insurer consists of polluting borrowing-firm loans (i.e., brown loans), green borrowing-firm loans (i.e., green loans), and affiliated reinsurance (reinsurance loans). We assume the ceding insurer's underlying asset portfolio's market value, $V_{ci}$, varies over the period continuously and according to the stochastic process:

$$dV_{ci}/V_{ci} = \mu_{ci}dt + \sigma_{ci}dW_{ci} \tag{9}$$

where

$$V_{ci} = (1 + R_g)\theta A + (1 + R_c)(1 - \theta)A + (1 + R_{ci})\beta L$$

$R_g$ = the green loan market rate, $R_c$ = the brown loan market rate

$\mu_{ci}$ = the instantaneous drift, $\sigma_{ci}$ = the instantaneous volatility

$W_{ci}$ = a Wiener process

It is worthy of mentioning that the instantaneous drift and volatility implicitly include the war impact. The implicit treatment results from the capped credit risks from borrowing-firm investment, as shown in Eqs (5) and (6).

Reinsurance is the ceding insurer's guarantee and the affiliate's liabilities directly proportional to payments to policyholders. We express the ceding insurer's net debt's book value, $Z = Le^R - \beta Le^{R_{ci}}$, with its maturity at the end of the period. The reinsurance cost burden for the ceding insurer is $(R - R_{ci}) > 0$, where $R$ and $R_{ci}$ are the guaranteed rate of the life insurance policy and the reinsurance, respectively. The net debt is the call option's strike price since the equity's market value is a call on the underlying asset portfolio.

By integrating Briys and de Varenne [13] with Dermine and Lajeri [28], a capped call option for the ceding insurer's equity value can be obtained by applying the risk-neutral valuation method:

$$S_{ci} = C_{ci}(M_h + M_l, Z) - \delta C_{ci}(\alpha(M_h + M_l), Z) \tag{10}$$

where $\delta$ = the life insurance policy's participation rate. The equity consists of two European capped calls with the same strike price. The first specification is:

$$C_{ci}(M_h + M_l, Z) = [(M_h + M_l)N(c_1) - Ze^{-R_B}N(c_2)]$$

$$- [(M_h + M_l)N(c_3) - V_{ci}e^{-R_B}N(c_4)] \tag{11}$$

where

$R_B$ = the risk-free rate

$N(\cdot)$ = the cumulative distribution function of the standard normal distribution

$$c_1 = \frac{1}{\sigma_{ci}}(\ln\frac{M_h + M_l}{Z} + R_B + \frac{\sigma_{ci}^2}{2}), \ c_2 = c_1 - \sigma_{ci}$$

$$c_3 = \frac{1}{\sigma_{ci}}(\ln\frac{M_h + M_l}{V_{ci}} + R_B + \frac{\sigma_{ci}^2}{2}), \ c_4 = c_3 - \sigma_{ci}$$

$$\sigma_{ci} = [\sigma_h + (b_h + b_h^2/2) + \sigma_l + (b_l + b_l^2/2)]/2$$

We assume that the volatility of the ceding insurer's asset portfolio return, $\sigma_{ci}$, is an average deviation of the two borrowing-firm assets returns. It makes the borrowing-firm credit risks can be a definitive treatment in the valuation of the ceding insurer's equity. Analogously, we can have the second specification:

$$C_{ci}(\alpha(M_h + M_l), Z) = [\alpha(M_h + M_l)N(c_5) - Ze^{-R_B}N(c_6)] - [\alpha(M_h + M_l)N(c_7) - V_{ci}e^{-R_B}N(c_8)] \tag{12}$$

where

$$c_5 = \frac{1}{\sigma_{ci}}(\ln\frac{\alpha(M_h + M_l)}{Z} + R_B + \frac{\sigma_{ci}^2}{2}), \ c_6 = c_5 - \sigma_{ci}$$

$$c_7 = \frac{1}{\sigma_{ci}}(\ln\frac{\alpha(M_h + M_l)}{V_{ci}} + R_B + \frac{\sigma_{ci}^2}{2}), \ c_8 = c_7 - \sigma_{ci}$$

Eq (11) is a limited-liability capped call, while Eq (12) is a participation-clause capped call belonging to life insurance policyholders. In both Eqs (11) and (12), the second terms of the right-hand side, are the value losses caused by the cap. The standard call reduced the above two terms.

Similarly, the affiliate's equity is a hybrid position by applying Briys and de Varenne [13]:

$$S_a = C_a(V_a, Z_a) - \delta C_a(\alpha_a V_a, Z_a) \tag{13}$$

The underlying assets' market value of the affiliate $V_a = (1+R_a)A_a$ follows a geometric Brownian motion such that:

$$dV_a / V_a = \mu_a dt + \sigma_a dW_a$$

where definitions of the parameters are analogous to those in Eq (9). For simplicity, we assume the two conditions ($\mu_a = \mu_{ci}$ and $\sigma_a = \sigma_{ci}$) because the ceding insurer and its affiliate belong to the same insurance holding company. The affiliate's liabilities (i.e., the strike price $Z_a$) are $\beta L e^{R_{ci}}$. The strike price demonstrates the affiliate reinsurance operations within the insurance holding company. Thus, the call positions are:

$$C_a(V_a, Z_a) = V_a N(d_9) - Z_a e^{-R_B} N(d_{10}) \tag{14}$$

$$C_a(\alpha_a V_a, Z_a) = \alpha_a V_a N(d_{11}) - Z_a e^{-R_B} N(d_{12}) \tag{15}$$

where

$$d_9 = \frac{1}{\sigma_a}(\ln\frac{V_a}{Z_a} + R_B + \frac{\sigma_a^2}{2}), \ d_{10} = d_9 - \sigma_a$$

$$d_{11} = \frac{1}{\sigma_a}(\ln\frac{\alpha_a V_a}{Z_a} + R_B + \frac{\sigma_a^2}{2}), \ d_{12} = d_{11} - \sigma_a$$

Eqs (14) and (15) are the European calls, not capped calls, as shown in Eqs (10) and (11), with the same strike price since we do not double-count the credit risks of the borrowing firms involved in the cap-and-trade mechanism.

## 3.3 Objective function and liabilities

In light of the previous green finance, we assume the ceding insurer choose the optimal guaranteed rate to maximize the insurance holding company's market value, which is defined as the equity values of ceding insurer and its affiliate:

$$\underset{R}{Max} \ S = S_{ci} + S_a \tag{16}$$

With information about the objective, we can evaluate the insurance holding company's liability by applying Dermine and Lajeri [28]:

$$PUT_{lia} = PUT_{ci} + PUT_a \tag{17}$$

where

$$PUT_{ci} = Ze^{-R_B} - [Ze^{-R_B}(1 - N(d_2)) - (V_h + V_l)(1 - N(d_1))] + \delta C_{ci}(\alpha(V_h + V_l), Z)$$

$$PUT_a = Z_a e^{-R_B} - [Z_a e^{-R_B}(1 - N(d_{10})) - V_a(1 - N(d_9))] + \delta C_{ci}(\alpha V_a, Z_a)$$

The term $PUT_{ci}$ is the ceding insurer's liability, similar to the put option discussed in the literature, except the underlying asset is the borrowing-firm assets, not the insurer's asset portfolio [28]. The term $PUT_a$ is the affiliate's liability. The same pattern applies.

## 4. Comparative static analyses

Differentiating Eq (16) with respect to the guaranteed rate, the optimal rate satisfies the first-order condition

$$\frac{\partial S}{\partial R} = \left(\frac{\partial C_{ci}(V_h + V_l, Z)}{\partial R} - \delta\frac{\partial C_{ci}(\alpha(V_h + V_l), Z)}{\partial R}\right)$$

$$+\left(\frac{\partial C_a(V_a, Z_a)}{\partial R} - \delta\frac{\partial C_a(\alpha_a V_a, Z_a)}{\partial R}\right) = 0 \tag{18}$$

The second-order condition ($\partial^2 S/\partial R^2 < 0$) validates the insurance holding company's equity maximization. The optimal guaranteed rate determines the marginal value of limited-liability calls equal to that of marginal participation-clause calls.

It is necessary to elaborate on the comparative statics about. We define the optimal insurer interest margin as the spread between the market liquid-asset rate and the optimal guaranteed rate before proceeding with the comparative static analyses regarding green finance, reinsurance, and war impact. The margin conveys information on the efficiency of financial intermediation, such as an efficient asset-liability matching management [31].

We first consider the effect on the ceding insurer's interest margin, from changes in green loans ($\theta$), the regulatory cap of the cap-and-trade scheme ($c_{cap}$), the affiliated reinsurance ($\beta$), and the structural break in return and volatility of the polluting borrowing-firm ($b_h$) brought from the war impact. Implicit differentiation of Eq (18) to parameter $i$ yields:

$$\frac{\partial R}{\partial i} = -\frac{\partial^2 S}{\partial R \partial i}\Big/\frac{\partial^2 R}{\partial R^2} \ \forall \ i = \theta, \ c_{cap}, \ \beta, \ \text{and} \ b_h \tag{19}$$

In addition, we can also investigate the policyholder protection issue. Deafferenting Eq (17) evaluated at the optimal guaranteed rate for the parameter $i$:

$$\frac{dPUT_{lia}}{di} = \frac{\partial PUT_{lia}}{\partial i} + \frac{\partial PUT_{lia}}{\partial R}\frac{\partial R}{\partial i} \ \forall \ i = \theta, \ c_{cap}, \ \beta, \ \text{and} \ b_h \tag{20}$$

The first term on the right-hand side of Eq (19) is the direct effect, holding the optimal guaranteed rate constant, the effect of the parameter on policyholder protection. The second term is the indirect effect: the effect of a parameter on the policyholder protection through the adjustments of every possible optimal state. We use numerical analysis to explain the intuition of Eqs (19) and (20) in the following section.

## 5. Methodology and data

Our model is a one-period (i.e., one-year) one. The numerical analysis critically relies on the firm-level data collected. Toward that end, we apply the method used by Chen et al. [15], Chen et al. [16], Huang et al. [17], and Li et al. [18]. Mainly, the firm-level parameters we collect for the numerical exercise are from different studies, and some are from assumptions based on theoretical reasonings. It is well-recognized that other data selections yield different quantitative results, but the qualitative ones should remain the same.

A numerical method is a mathematical way to solve particular problems such as Eqs (19) and (20) in our model. We may have a significant advantage of numerical analysis to solve the

optimal guaranteed rate and the comparative statics. An additional benefit is that a numerical method only uses the evaluation of standard functions and the operations developed in our model. We conduct the numerical analysis based on the baseline as follows:

i. Supply loci: We assume that the ceding insurer faces an upward-sloping life insurance policy curve [9]. Besides, Frailich and Metz [32] have reviewed several dozen life insurance policies and report that guaranteed rates are often 1% to 2%. Both the assumption and the report allow us to express the policy supply loci as $(R(\%),L)$ = (1.00, 307), (1.20, 323), (1.40, 334), (1.60, 341), (1.80, 345), (2.00, 347), and (2.20, 348) where the firm-level life insurance policy quantity in each bundle is arbitrary.

ii. Affiliated reinsurance: Koijen and Yogo [4] use U.S. data on reinsurance agreements and document that life insurers cede 25% of every dollar insured to shadow reinsurers in the year 2012. Accordingly, we assume $\beta$ = 25%.

iii. Ceding insurer's capital ratio: We assume that the capital ratio, $1-\alpha$, equals 10%. Given a liability level ($L$ = 345), we have $K_g$ = 38.33. Thus, we have $(\theta A_g + (1-\theta)A_c)$ = 297.08. Liu [30] reports that the proportion of green loans relative to total loans exceeds 10%. We here assume that $\theta A_g/(\theta A_g + (1-\theta)A_c)$ = 12%. We then obtain $A_g$ = 73.30 and $A_c$ = 520.86 when $\theta$ = 0.50.

iv. Affiliated reinsurer: We take the conditions of $K_a = K_g$ = 38.33 and $(1-\kappa) = \beta$ = 25% for simplicity. Accordingly, we have the amount of $A_a$ = 345×25%+38.33×25% = 95.83.

v. High-carbon emitter: We follow Dermine and Lajeri [28] to assume the capital-to-liability ratio to be 30%. Financial firms generally have a lower level of capital-to-liability ratio than productive firms. Thus, we have $K_h$ = 30%×$(1-\theta)A_c$ = 78.13. According to Eq (1), $A_h$ = 260.43+78.13 = 338.56.

vi. Low-carbon emitter: One effort to efficiently reduce carbon emissions in production is through green technology investment. Low-carbon emitters are generally more willing to have capital investment to reduce carbon emissions than high-carbon emitters. We apply Dermine and Lajeri [28] to specify the high-emitter capital-to-liability ratio to be 40% since the low-emitter one is 30%, as mentioned above. Thus, we obtain $K_l$ = 40%×$\theta A_g$ = 14.66. As shown in Eq (2), we can have $A_l$ = 51.31.

vii. Investment returns: The yearly interest bond rate is 2.41% (https://www.bloomberg.com/markets/rates-bonds/government-bonds/us). Tan et al. [33] find that the investment return rate is about 5.10%. Accordingly, the green loan rate $R_g$ is assumed to be 4.60%, while the brown loan rate is $R_c$ = 5.10%. Liu [30] argues that green loans are less risky than brown loans. We also assume that the high-carbon-emitter investment return rate is $R_h$ = 5.60%, and the low-carbon-emitter investment return rate is $R_l$ = 5.10% as a baseline for our analysis.

viii. Risks: With reference to Brockman and Turtle [12], we assume $\sigma_h$ = 29.04%. In addition, we presume $\sigma_l$ = 24.04%, as previously argued by Liu [30]. For simplicity, the condition of $\sigma_{ci} = \sigma_l$ because of affiliated reinsurance. According to Tan et al. [33], we consider the structural break at $b_h = b_l$ = 3.00% capturing the war impact. The participation level of the life insurance policy is $\delta$ = 85% [13].

ix. Cap-and-trade scheme: Narassimhan et al. [34] report that the cost of compliance with the economic efficiency of the cap-and-trade system is $72,440 per installation, the administration cost is $2,750, and the stringency of the cap (% cap reduction/year) is 2.20%. Based on

the three statistics above, we assume $c_h$ = 2,750/72,440 = 3.80%, $c_{cap}$ = 2.20%, and $c_l$ = 0.60%. Our assumption implies a carbon-neutral case.

## 6. Results

Climate finance provides financings to borrowing firms to cope with climate change adaptation and mitigation. Green finance is broader in scope and covers a wide spectrum of the environment, including biodiversity conservation and restoration. Our comparative statics include but do not distinguish the issue of climate and green finances. First, we examine the impact of the green lending of the insurer on its guaranteed rate.

Table 1 shows that the insurer's green lending increases the guaranteed rate; the stringent regulatory cap of the cap-and-trade mechanism reinforces this effect. The insurer needs more funds to increase green lending by increasing life policy policies at an increased guaranteed rate, which implies the insurer's interest margin reduction. The stringent regulatory cap makes the insurer's interest margin reduce more significant. Our result demonstrates that the green finance provided by the insurer helps the green borrowing firm to have a cleaner production for environmental improvement. But, the improved greenness costs the interest margin of the insurer. Under the circumstances, the insurer may be reluctant to provide green loans unless, for example, the government subsidizes the insurer's green finance toward sustainability. Yang et al. [25] suggest that government subsidy is an indispensable incentive for greenness improvement. Similarly, our result is consistent with their suggestion from the insurer's green finance perspective, particularly when the regulatory cap becomes stringent.

In addition, we find that increasing green loans increases the insurer's policyholder protection, and the stringent regulatory cap reinforces this positive impact. Two effects explain our findings. The direct effect indicates that increasing the insurer's green loans increases policyholder protection, holding the guaranteed rate constant. Green loans, reconsidering investment policy related to climate, involve an environmental and sustainability topic into a core

**Table 1. Responsiveness of the optimal guaranteed rate and the insurer's liabilities ($PUT_{lia}$) to green loans ($\theta$) at various levels of $c_{cap}$.**

|  | $c_{cap}$ |  |  |  |  |  |  |
|---|---|---|---|---|---|---|---|
| $\theta$ | 0.0145 | 0.0170 | 0.0195 | 0.0220 | 0.0245 | 0.0270 | 0.0295 |
|  | $\partial R/\partial\theta$(%) |  |  |  |  |  |  |
| 0.35→0.40 | 1.0277 | 1.0073 | 0.9870 | 0.9669 | 0.9469 | 0.9270 | 0.9072 |
| 0.40→0.45 | 1.0175 | 0.9972 | 0.9770 | 0.9569 | 0.9370 | 0.9171 | 0.8975 |
| 0.45→0.50 | 1.0073 | 0.9871 | 0.9670 | 0.9470 | 0.9271 | 0.9074 | 0.8878 |
| 0.50→0.55 | 0.9972 | 0.9770 | 0.9570 | 0.9370 | 0.9173 | 0.8976 | 0.8781 |
| 0.55→0.60 | 0.9871 | 0.9670 | 0.9470 | 0.9272 | 0.9075 | 0.8879 | 0.8684 |
| 0.60→0.65 | 0.9771 | 0.9571 | 0.9371 | 0.9174 | 0.8977 | 0.8782 | 0.8588 |
|  | $dPUT_{lia}/d\theta$:total effect |  |  |  |  |  |  |
| 0.35→0.40 | 2.7260 | 2.7021 | 2.6781 | 2.6542 | 2.6304 | 2.6066 | 2.5828 |
| 0.40→0.45 | 2.7141 | 2.6902 | 2.6663 | 2.6424 | 2.6185 | 2.5948 | 2.5710 |
| 0.45→0.50 | 2.7022 | 2.6783 | 2.6544 | 2.6305 | 2.6067 | 2.5829 | 2.5592 |
| 0.50→0.55 | 2.6903 | 2.6664 | 2.6425 | 2.6187 | 2.5949 | 2.5711 | 2.5474 |
| 0.55→0.60 | 2.6784 | 2.6545 | 2.6306 | 2.6068 | 2.5830 | 2.5593 | 2.5356 |
| 0.60→0.65 | 2.6665 | 2.6426 | 2.6188 | 2.5950 | 2.5712 | 2.5475 | 2.5238 |

Notes: Unless otherwise indicated, see Table 2. The shaded areas illustrate the optimal values and the comparative statics evaluated at the optimal guaranteed rate ($R$(%), $L$) = (2.00, 347). In the lower panel, the direct and indirect effects are positive.

**Table 2. Data description.**

| Variable | approximation and assumption | source |
|---|---|---|
| ($R(\%)L$): policy supply | (1.00, 307), (1.20, 323), (1.40, 334), (1.60, 341), (1.80, 345), (2.00, 347), (2.20, 348) | [32] |
| $\alpha$: leverage | 90% | [13] |
| $\beta$: proportion of reinsurance from the insurer | 25% | [4] |
| $\theta$: proportion of green-linked loans | 50% | [30] |
| $K_h$: pollutig borrowing-firm capital | 78.13 | [28] |
| $K_l$: green borrowing-firm capital | 14.66 | |
| $\alpha_a$: the affiliate's leverage ratio | 90% | [13] |
| $\kappa$: capital regulation | 75% | |
| $R_B$: risk-free interest rate | 2.41% | Bloomberg |
| $R_g$: green loan rate | 4.60% | [33] |
| $R_c$: brown loan rate | 5.10% | [33] |
| $R_h$: polluting borrowing-firm investment return rate | 5.60% | [30] |
| $R_l$: green borrowing-firm investment return rate | 5.10% | [30] |
| $\sigma_h$: polluting borrowing-firm volatility | 29.04% | [12] |
| $\sigma_l$: green borrowing-firm volatility | 24.04% | [30] |
| $b_h$: polluting borrowing-firm structural break | 3.00% | [33] |
| $b_l$: green borrowing-firm volatility structural break | 3.00% | [33] |
| $\delta$: participation level | 85% | [13] |
| $c_h$: exceeding carbon emission allowance cost | 3.80% | [34] |
| $c_{cap}$: regulatory cap | 2.20% | [34] |
| $c_l$: benefit from selling carbon emission allowance | 0.60% | [34] |
| $R_{ci}$: reinsurance guaranteed rate | 0.5% | |
| $R_a$: investment return rate of the reinsurer | 5.6% | |
| $\sigma_a$: volatility of the reinsurer | 29.04% | |

Sources and explanations: see (i)~(ix) in Section 5.

business model. The green involvement earns goodwill from customer greenness awareness [35] and enhances policyholder protection. The indirect effect suggests that the insurer's green loans improve the life insurance policies at an increased guaranteed rate. Then, this increased guaranteed rate leads to higher policyholder protection. Considering the optimal guaranteed rate adjustments, we have a positive relationship between green loans and policyholder protection. The indirect effect reinforces an overall positive impact that green loans the insurer provides help policyholder protection, affecting insurance stability. Chiaramonte et al. [5] suggest that sustainability expressed by ESG scores enhances the stability of insurance. Our results contribute to the insurance literature that green loans toward sustainability improve insurance stability but at the reduced insurer interest margin cost. The margin reduction may encourage the government to subsidize the insurer's sustainability practices (i.e., green loans in our model) for its borrowing-firm cleaner production for environmental improvement, mainly when the regulatory cap is strict.

**Table 3. Responsiveness of the optimal guaranteed rate and the insurer's liabilities ($PUT_{lia}$) to $c_{cap}$ at various levels of green loans ($\theta$).**

| $c_{cap}$ | $\theta$ | | | | | | |
|---|---|---|---|---|---|---|---|
| | 0.35 | 0.4 | 0.45 | 0.5 | 0.55 | 0.6 | 0.65 |
| | $\partial R/\partial c_{cap}$ | | | | | | |
| 0.0145→0.0170 | 0.5880 | 0.5840 | 0.5799 | 0.5640 | 0.5719 | 0.5679 | 0.5640 |
| 0.0170→0.0195 | 0.5798 | 0.5758 | 0.5718 | 0.5560 | 0.5638 | 0.5599 | 0.5560 |
| 0.0195→0.0220 | 0.5717 | 0.5677 | 0.5637 | 0.5480 | 0.5558 | 0.5519 | 0.5480 |
| 0.0220→0.0245 | 0.5636 | 0.5596 | 0.5556 | 0.5400 | 0.5478 | 0.5439 | 0.5400 |
| 0.0245→0.0270 | 0.5555 | 0.5516 | 0.5476 | 0.5321 | 0.5399 | 0.5360 | 0.5321 |
| 0.0270→0.0295 | 0.5475 | 0.5436 | 0.5397 | 0.5243 | 0.5320 | 0.5281 | 0.5243 |
| | $dPUT_{lia}/dc_{cap}$: total effect | | | | | | |
| 0.0145→0.0170 | 154.6867 | 154.2089 | 153.7313 | 151.8246 | 152.7773 | 152.3007 | 151.8246 |
| 0.0170→0.0195 | 153.7272 | 153.2502 | 152.7734 | 150.8700 | 151.8210 | 151.3453 | 150.8700 |
| 0.0195→0.0220 | 152.7696 | 152.2933 | 151.8174 | 149.9174 | 150.8667 | 150.3918 | 149.9174 |
| 0.0220→0.0245 | 151.8138 | 151.3384 | 150.8633 | 148.9668 | 149.9143 | 149.4403 | 148.9668 |
| 0.0245→0.0270 | 150.8599 | 150.3854 | 149.9112 | 148.0183 | 148.9639 | 148.4909 | 148.0183 |
| 0.0270→0.0295 | 149.9080 | 149.4344 | 148.9611 | 147.0718 | 148.0156 | 147.5435 | 147.0718 |

Note: Unless otherwise indicated, the same as Table 1.

The government usually shrinks the regulatory cap for carbon emission reductions (Du et al., 2016). The result in Table 3 demonstrates that the shrunk regulatory cap decreases the life insurance policies at a reduced guaranteed rate; the former lead to an increase in insurer interest margin. The increased margin becomes more significant when the insurer's green lending decreases. When the government decides to lower the regulatory cap, the green borrowing firm reduces the revenues from selling carbon allowances. The polluting borrowing firm raises the costs of buying carbon allowance. The reduced cap policy may incentivize the firms to increase their efficiency of cleaner production. Therefore, we can conclude that green finance makes the insurer enhance profits and accelerate cleaner production toward environmental sustainability.

In Table 3, the positive direct effect demonstrates that decreasing the regulatory cap decreases policyholder protection, holding the optimal guaranteed rate constant. A decrease in regulatory cap makes borrowing firms under the cap-and-trade mechanism increase cleaner production, yielding the green borrowing-firm carbon allowance revenue reduction and the polluting-borrowing carbon allowance cost reinforcement. Overall, the cleaner production costs borrowing firms, which reduces the investment incentives and then decreases the funds needed from the insurer. Thus, the derived demand for funds reduces the insurer's liabilities that harm policyholder protection. The indirect effect is also positive in sign. As mentioned previously, a stringent regulatory cap in the cap-and-trade mechanism decreases the insurer's guarantee rate, further decreasing the insurer's liabilities. Thus, the rigorous regulatory cap harms policyholder protection through every possible state of the optimal guaranteed rate adjustments. The indirect effect reinforces the direct impact to give an overall positive effect: a stringent cap deteriorates policyholder protection, adversely affecting insurance stability, particularly when the insurer's green lending is low.

Table 4 shows that increasing affiliated reinsurance increases the optimal guaranteed rate. The result is very straightforward. Increasing the affiliated reinsurance requires more funds by increasing life insurance policies and raising the guaranteed rate. However, a higher guaranteed rate implies the insurer's interest margin reduction. The reinsurance costs the insurer from the viewpoint of insurance profitability.

**Table 4. Responsiveness of the optimal guaranteed rate and the insurer's liabilities ($PUT_{lia}$) to affiliated reinsurance ($\beta$).**

| $\beta$ | (R(%),L) | | | | | | |
|---|---|---|---|---|---|---|---|
| | (1.00, 307) | (1.20, 323) | (1.40, 334) | (1.60, 341) | (1.80, 345) | (2.00, 347) | (2.20, 348) |
| | $\partial R/\partial\beta$ | | | | | | |
| 0.175→0.200 | - | 1.9401 | 1.6917 | 1.4598 | 1.2803 | 1.3277 | - |
| 0.200→0.225 | - | 1.7306 | 1.5078 | 1.3009 | 1.1419 | 1.1881 | - |
| 0.225→0.250 | - | 1.5447 | 1.3449 | 1.1603 | 1.0197 | 1.0652 | - |
| 0.250→0.275 | - | 1.3771 | 1.1983 | 1.0340 | 0.9102 | 0.9553 | - |
| 0.275→0.300 | - | 1.2240 | 1.0645 | 0.9190 | 0.8107 | 0.8556 | - |
| 0.300→0.325 | - | 1.0824 | 0.9410 | 0.8130 | 0.7192 | 0.7643 | - |
| | $dPUT_{lia}/d\beta$:total effect | | | | | | |
| 0.175→0.200 | - | 345.0340 | 280.2902 | 236.5723 | 209.9396 | 197.1580 | - |
| 0.200→0.225 | - | 321.2564 | 263.0405 | 223.8065 | 199.9382 | 188.5042 | - |
| 0.225→0.250 | - | 299.0383 | 246.6763 | 211.4556 | 190.0562 | 179.8230 | - |
| 0.250→0.275 | - | 278.0022 | 230.9766 | 199.4076 | 180.2501 | 171.1053 | - |
| 0.275→0.300 | - | 257.8837 | 215.7927 | 187.5937 | 170.5010 | 162.3569 | - |
| 0.300→0.325 | - | 238.5008 | 201.0285 | 175.9773 | 160.8092 | 153.5963 | - |

Note: Unless otherwise indicated, the same as Table 1.

We examine the effect of affiliate reinsurance on policyholder protection and present the results in Table 4. The direct impact shows the positive effect of affiliate reinsurance on the policyholder protection, holding the guaranteed rate constant. Certainly, policyholders need reinsurance for more security. The indirect impact indicates, as mentioned previously, that increasing affiliate reinsurance increases the guaranteed rate. This increased guaranteed rate further leads to securer policyholder protection. The indirect and direct effects reinforce each other, yielding an overall positive result: increasing affiliate reinsurance helps policyholder protection.

The uncertainty of war has tangentially generated outcomes that will arguably lead to promoting and impeding businesses. Table 5 examines how uncertainty brought by a war impact the insurer's guaranteed rate-setting behavior. In our model, structural breaks in mean and volatility of the return faced by the polluting borrowing-firm capture the war impact. We show that the significant seriousness of the war's impact reduces the life insurance policies' guaranteed rate. The war impact makes the insurer more conservative in extending the life insurance businesses. The conservation manifests the insurer with reduced guaranteed rate-setting behavior, increasing the insurer's interest margin. Therefore, the war impact helps the conservative insurer from a profit perspective.

Table 5 also demonstrates that increasing the war impact harms policyholder protection. The direct effect indicates economic intuitively, holding the optimal guaranteed rate constant, the war impacts the polluting borrowing firm decreases policyholder protection significantly. Due to the considerable war impact, the insurer is reluctant to provide life insurance policies. The indirect effect shows that a war impacting the polluting borrowing firm yields a reduced guaranteed rate. It makes the policyholder less protected through every possible state of optimal guaranteed rate adjustments. The indirect and direct effects reinforce each other, and they have an overall negative impact. We can conclude that the war's impact on the polluting borrowing firm hurts policyholder protection, adversely affecting insurance stability. One immediate implication is that the war impedes the polluting borrowing firm from participating in

**Table 5. Responsiveness of the optimal guaranteed rate and the insurer's liabilities ($PUT_{lia}$) to the war impact ($b_h$).**

| $b_h$ | (R(%),L) | | | | | | |
|---|---|---|---|---|---|---|---|
| | (1.00, 307) | (1.20, 323) | (1.40, 334) | (1.60, 341) | (1.80, 345) | (2.00, 347) | (2.20, 348) |
| | $\partial R/\partial b_h$ | | | | | | |
| 0.015→0.020 | - | -0.7028 | -0.6112 | -0.5250 | -0.4579 | −0.4721 | - |
| 0.020→0.025 | - | -0.6990 | -0.6073 | -0.5209 | -0.4534 | −0.4653 | - |
| 0.025→0.030 | - | -0.6951 | -0.6033 | -0.5168 | -0.4488 | −0.4584 | - |
| 0.030→0.035 | - | -0.6912 | -0.5993 | -0.5127 | -0.4443 | −0.4515 | - |
| 0.035→0.040 | - | -0.6872 | -0.5953 | -0.5086 | -0.4397 | −0.4446 | - |
| 0.040→0.045 | - | -0.6832 | -0.5912 | -0.5044 | -0.4350 | −0.4377 | - |
| | $dPUT_{lia}/db_h$:total effect | | | | | | |
| 0.015→0.020 | - | -117.4253 | -93.5373 | -77.4076 | -67.6301 | −62.9566 | - |
| 0.020→0.025 | - | -117.3752 | -93.6224 | -77.5991 | -67.8929 | −63.2512 | - |
| 0.025→0.030 | - | -117.3231 | -93.7079 | -77.7925 | -68.1582 | −63.5486 | - |
| 0.030→0.035 | - | -117.2691 | -93.7941 | -77.9878 | -68.4261 | −63.8490 | - |
| 0.035→0.040 | - | -117.2137 | -93.8810 | -78.1852 | -68.6967 | −64.1522 | - |
| 0.040→0.045 | - | -117.1570 | -93.9689 | -78.3846 | -68.9699 | −64.4583 | - |

Note: Unless otherwise indicated, the same as Table 1. In the lower panel, the direct and indirect effects are adverse.

the cap-and-trade scheme for cleaner production toward environmental sustainability and further hurts the fund provider's policyholder protection, contributing to insurance instability.

# 7. Conclusion and policy implications

We have developed a capped call option to explore the effects of insurer green lending, the regulatory cap of the cap-and-trade mechanism, affiliated reinsurance, and a war on the insurer's performance. We show that increasing green loans toward borrowing-firm cleaner production decreases the insurer's interest margin. The regulatory reduced cap raises the insurer's margin, harming policyholder protection. The affiliated reinsurance reduces the insurer's margin but benefits policyholder protection. Finally, increasing the war impact on the polluting borrowing firm makes the insurer more conservative by decreasing the insurance businesses. Our results should be attractive to investors, managers, and regulators.

As a market-oriented policy for controlling carbon emissions, the cap-and-trade mechanism brought new challenges to insurer green finance decisions, leading to cleaner production toward environmental sustainability. To face the challenges, we present two essential suggestions. We suggest the government green subsidy to stimulate the insurer to increase green lending toward environmental sustainability without hurting insurance stability. The government primarily focusing on decarbonization should also pay sufficient attention to ensuring climate and green finances, yielding financial stability. Therefore, our research complements the literature by providing two regulatory suggestions about sustainability practices, insurer operations, and insurance stability.

Our framework is reasonably general and should open several further avenues of research. For example, an outgrowth of the model is to introduce the link between the borrowing-firm cleaner production planning and the cap-and-trade mechanism. This link then can allow the model to analyze the effect of the borrowing-firm cleaner production plans on the financial institution's green finance decisions toward sustainable development. Besides, the model can accommodate borrowing-firm technology choices. The model can investigate the impact of

cleaner production related to corporate sustainability and social responsibility on insurer green finance.

## Supporting information

**S1 File. Simulation.**
(XLSX)

**S2 File. Computation of tables.**
(DOCX)

## Acknowledgments

The authors would like to thank the Academic Editor and two anonymous referees for their helpful comments and suggestions. The usual disclaimer applies.

## Author Contributions

**Conceptualization:** Jyh-Jiuan Lin.

**Data curation:** Jyh-Jiuan Lin.

**Formal analysis:** Jyh-Jiuan Lin.

**Funding acquisition:** Jyh-Jiuan Lin.

**Investigation:** Jyh-Jiuan Lin.

**Methodology:** Jyh-Jiuan Lin.

**Project administration:** Jyh-Jiuan Lin.

**Resources:** Fu-Wei Huang.

**Software:** Fu-Wei Huang.

**Supervision:** Fu-Wei Huang.

**Validation:** Fu-Wei Huang.

**Visualization:** Fu-Wei Huang.

**Writing – original draft:** Fu-Wei Huang.

**Writing – review & editing:** Fu-Wei Huang.

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
