## [Decision Letter · Decision Letter 0]

14 Dec 2022

PONE-D-22-29807Insurer green finance under regulatory cap-and-trade mechanism associated with green/polluting production during a warPLOS ONE

Dear Dr. Lin,

Thank you for submitting your manuscript to PLOS ONE. After careful consideration, we feel that it has merit but does not fully meet PLOS ONE’s publication criteria as it currently stands. Therefore, we invite you to submit a revised version of the manuscript that addresses the points raised during the review process.

ACADEMIC EDITOR:

The reviewers are experts in the area and have prepared a careful and fair review. They both recommended the major revision, in which I agree with. I would suggest you to carefully follow the comments to revise your manuscript and resubmit it for re-consideration for publication. For publication, you must address at least the following summary of key points as below. However, please provide full responses to all comments of two reviewers. Please find detailed comments in the review reports.

1. Introduction- clearly state the context and main findings of the paper with main references (articles) you are contributing to the stream of related literature.

2. Literature - could you please provide one separate part to explain 'the cap-and-trade' mechanism for audiences to be more simpler in understanding this policy/mechanism.

3. Data and methods - could you please clearly stay data vendor with main methods used for the study and 'why's (could add some related papers to support your used methods). Please make the methodological parts concise highlighting the main ones.

4. Findings - please highlight the main findings and explain how you contribute further to the literature (references you review in the introduction and literature review). - Do this as best as you can but we do not require the paper published in PLOS ONE to show their contribution so I would leave it to your choice.

5. Conclusion - could you please provide one paragraph for main context, one paragraph for main findings and one for future direction and policy.

6. Though the paper in general is written in understandable English some minor typos and grammatical errors need to be corrected.

7. It is not clear why you employ a contingent claim model. What are the advantages compared to alternative models?

8. They assume in (vi) a capital-to-liability ratio to be 40% (p. 19) but no specific reason is given for this percentage. You should give a better explanation for this.

9.You also consider in (vii) a bond interest rate of 2.41%. Is this an annual rate?

10. Finally, the sample is poorly described. What is the period? When does it begin? And when does it end? How many observations were gathered? Also, the main summary statistics are missing as well as a normality test.

And other minor comments from TWO reviewers.

We look forward to receiving your revised manuscript.

Kind regards,

Vu Quang Trinh, PhD

Academic Editor

PLOS ONE

Journal Requirements:

Additional Editor Comments:

Both reviewers think that the statistical analysis has NOT yet been performed appropriately and rigorously. This is very important so I would suggest you addressing this properly.

Reviewers' comments:

Reviewer's Responses to Questions

**Comments to the Author**

1. Is the manuscript technically sound, and do the data support the conclusions?

Reviewer #1: Yes

Reviewer #2: No

2. Has the statistical analysis been performed appropriately and rigorously? 

Reviewer #1: No

Reviewer #2: No

3. Have the authors made all data underlying the findings in their manuscript fully available?

Reviewer #1: No

Reviewer #2: Yes

4. Is the manuscript presented in an intelligible fashion and written in standard English?

Reviewer #1: Yes

Reviewer #2: No

5. Review Comments to the Author

Reviewer #1: This paper addresses the insurer green finance under regulatory cap-and-trade mechanism associated with

green/polluting production during a war. This is a very interesting subject not yet addressed in literature, therefore deserving further attention. Congratulations! However, it should be improved in order to meet PLOS ONE standards. Thus, my comments to the authors are as follows:

1. Though the paper in general is written in understandable English some minor typos and grammatical errors need to be corrected.

2. It is not clear why authors employ a contingent claim model. What are the advantages compared to alternative models?

3. They assume in (vi) a capital-to-liability ratio to be 40% (p. 19) but no specific reason is given for this percentage. Authors should give a better explanation for this.

4. They also consider in (vii) a bond interest rate of 2.41%. Is this an annual rate?

5. Finally, the sample is poorly describel. What is the period? When does it begin? And when does it end? How many observations were gathered? Also, the main summary statistics are missing as well as a normality test.

Therefore, my recommendation to the authors is to revise and resubmit.

Reviewer #2: Dear Authors,

My pleasure to review your paper. I appreciate your work and have some comments and thoughts for your revision (if possible).

In the introduction, the authors mentioned "this study aims to study the determinants of the life insurance policy’s optimal guaranteed rate setting and policyholder protection, considering credit risk from high-/low-carbon

emission borrowing firm" which is the main aim of this study. However, in the data and findings, I was not able to understand/see appropriate firm-level data to detect our main research.

I appreciate the way you elaborate data and mathematical explanations, however it is quite lengthy in my personal opinion, while the table the authors provide data sources and selected variables seems not clear to me for my best understanding.

I suggest the following points hoping the authors could consider for revision:

The general structure is as follows:

1. Introduction- clearly state the context and main findings of the paper with main references (articles) the authors are contributing to the stream of related literature.

2. Literature - could you please provide one seperate part to explain 'the cap-and-trade' mechanism for audiences to be more simpler in understanding this policy/mechanism.

3. Data and methods - could you please clearly stay data vendor with main methods used for ths study and 'why's (could add some related papers to support your used methods). Please make the methodological.parts concise highlighting the main ones

4. Findings - please highlight the main findings and explain how they contribute further to the literature (references you review in the introduction and literature review).

5. Conclusion - could you please provide one paragraph for main context, one paragraph for main findings and one for future direction and policy.

Hope my suggestions and comments are helpful for your revision.

I look forward to reading your revised work.

Regards,

Anonymous referee

6. PLOS authors have the option to publish the peer review history of their article (what does this mean?). If published, this will include your full peer review and any attached files.

Reviewer #1: No

Reviewer #2: No

---

## [Author Response · Author response to Decision Letter 0]

15 Jan 2023

Response to Reviewers

Ms. Ref. No.: PONE-D-22-29807

Title: Insurer green finance under regulatory cap-and-trade mechanism associated with green/polluting production during a war

PLOS ONE

Date: December 22, 2022

Point-by-point responses to the academic editor’s and reviewers’ comments

We sincerely appreciate the academic editor and reviewers for their insightful and constructive comments on our paper. Following their comments and suggestions, we have revised the article to address all their concerns. Below, we present point-by-point responses, quoting their comments and responding in the subsequent paragraphs.

Academic Editor:

“The reviewers are experts in the area and have prepared a careful and fair review. They both recommended the major revision, in which I agree with. I would suggest you to carefully follow the comments to revise your manuscript and resubmit it for re-consideration for publication. For publication, you must address at least the following summary of key points as below. However, please provide full responses to all comments of two reviewers. Please find detailed comments in the review reports.

[1-1]. Introduction- clearly state the context and main findings of the paper with main references (articles) you are contributing to the stream of related literature.

Response [1-1]:

Our four findings above demonstrate the influences of green loans, the regulatory cap of the cap-and-trade mechanism, affiliated reinsurance, and a war on the insurer’s performance in a borrowing-firm-insurer situation. Chen et al. (2022a) explore the impacts of the cap-and-trade transactions and financial gray rhino threats on insurer green performance. The research examines the effects of the government green subsidy with green trading on the insurer’s lending decisions (Chen et al., 2022b). Huang et al. (2022) analyze free riding and insurer carbon-linked investment. Li et al. (2023) investigate the green loan subsidy, regulatory cap, and green technology of borrowing-firm environmental impacts on insurer green finance assessment. Accordingly, our findings contribute to the literature on insurer green finance by extensively exploring the insurer’s green funding for the green and polluting borrowing firms participating in the cap-and-trade mechanism during a war.

(p. 6)

References

Chen, S., Huang, F.W., Lin, J.H., 2022a. Effects of cap-and-trade mechanism and financial gray rhino threats on insurer performance. Energies 15 (15), 5506. https://doi.org/10.3390/en15155506.

Chen, S., Huang, F.W., Lin, J.H., 2022b. Life insurance policyholder protection, government green subsidy, and cap-and-trade transactions in a black swan environment. Energy Econ. 115, 106333. https://doi.org/10.1016/j.eneco.2022.106333.

Huang, F.W., Chen, S., Lin, J.H., 2022. Free riding and insurer carbon-linked investment. Energy Econ. 17, 105838. https://doi.org/10.1016/j.eneco.2022.105838.

Li, X., Chen, L., Lin, J.H., 2023. Borrowing-firm environmental impact on insurer green finance assessment: Green loan subsidy, regulatory cap, and green technology. Environ. Impact Assess. Rev. 99, 107007. https://doi.org/10.1016/j.eiar.2022.107007.

The response is the same as Response [1-4].

[1-2]. Literature - could you please provide one separate part to explain ‘the cap-and-trade’ mechanism for audiences to be more simpler in understanding this policy/mechanism.

Response [1-2]:

Carbon emission transactions in the cap-and-trade scheme are a cost-oriented mechanism for carbon emission reductions. The government determines a cap to incentivize firms to reduce carbon emissions. Firms are carbon-allowance buyers required by the government that their carbon allowance emissions exceed the regulatory cap. The carbon emission buying for production permission costs the high emitters, which creates an incentive to get involved in reducing carbon emissions in production. The carbon emission selling benefits the low emitters since they have put efforts into reducing carbon emissions. Carbon buyers and sellers in cap-and-trade transactions may improve environmental development by reducing carbon emissions.

(p. 7)

[1-3]. Data and methods - could you please clearly stay data vendor with main methods used for the study and ‘why’s (could add some related papers to support your used methods). Please make the methodological parts concise highlighting the main ones.

Response [1-3]:

Our model is a one-period (i.e., one-year) one. The numerical analysis critically relies on the firm-level data collected. Toward that end, we apply the method used by Chen et al. (2022a), Chen et al. (2022b), Huang et al. (2022), and Li et al. (2023). Mainly, the firm-level parameters we collect for the numerical exercise are from different studies, and some are from assumptions based on theoretical reasonings. It is well-recognized that other data selections yield different quantitative results, but the qualitative ones should remain the same.

(p. 19)

[1-4]. Findings - please highlight the main findings and explain how you contribute further to the literature (references you review in the introduction and literature review). - Do this as best as you can but we do not require the paper published in PLOS ONE to show their contribution so I would leave it to your choice.

Response [1-4]:

The response is the same as Response [1-1].

Our four findings above demonstrate the influences of green loans, the regulatory cap of the cap-and-trade mechanism, affiliated reinsurance, and a war on the insurer’s performance in a borrowing-firm-insurer situation. Chen et al. (2022a) explore the impacts of the cap-and-trade transactions and financial gray rhino threats on insurer green performance. The research examines the effects of the government green subsidy with green trading on the insurer’s lending decisions (Chen et al., 2022b). Huang et al. (2022) analyze free riding and insurer carbon-linked investment. Li et al. (2023) investigate the green loan subsidy, regulatory cap, and green technology of borrowing-firm environmental impacts on insurer green finance assessment. Accordingly, our findings contribute to the literature on insurer green finance by extensively exploring the insurer’s green funding for the green and polluting borrowing firms participating in the cap-and-trade mechanism during a war.

(p. 6)

[1-5]. Conclusion - could you please provide one paragraph for main context, one paragraph for main findings and one for future direction and policy.

Response [1-5]:

We have developed a capped call option to explore the effects of insurer green lending, the regulatory cap of the cap-and-trade mechanism, affiliated reinsurance, and a war on the insurer’s performance. We show that increasing green loans toward borrowing-firm cleaner production decreases the insurer’s interest margin. The regulatory reduced cap raises the insurer’s margin, harming policyholder protection. The affiliated reinsurance reduces the insurer’s margin but benefits policyholder protection. Finally, increasing the war impact on the polluting borrowing firm makes the insurer more conservative by decreasing the insurance businesses. Our results should be attractive to investors, managers, and regulators.

As a market-oriented policy for controlling carbon emissions, the cap-and-trade mechanism brought new challenges to insurer green finance decisions, leading to cleaner production toward environmental sustainability. To face the challenges, we present two essential suggestions. We suggest the government green subsidy to stimulate the insurer to increase green lending toward environmental sustainability without hurting insurance stability. The government primarily focusing on decarbonization should also pay sufficient attention to ensuring climate and green finances, yielding financial stability. Therefore, our research complements the literature by providing two regulatory suggestions about sustainability practices, insurer operations, and insurance stability.

(pp. 28~29)

[1-6]. Though the paper in general is written in understandable English some minor typos and grammatical errors need to be corrected.

Response [1-6]:

Thank you. We have double-checked the paper.

[1-7]. It is not clear why you employ a contingent claim model. What are the advantages compared to alternative models?

Response [1-7]:

The contingent claims approach to the corporate securities valuation treats a firm’s equity as a call option on the firm’s assets (Black and Scholes, 1973) and a short put option on the firm’s value (Merton, 1973). Brockman and Turtle (2003) further formulate the corporate security valuation based on a path-dependent barrier option model, focusing on a premature structure. Briys and de Varenne (1994) use a contingent claim approach to evaluate a life insurance company’s equity. Grosen and Jørgensen (2002) apply Briys and de Varenne (1994) by introducing a premature state of an insurer’s equity valuation. This article develops the model in the spirit of Briys and de Varenne (1994), Brockman and Turtle (2003), and Grosen and Jørgensen (2002).

(p. 5)

References

Black, F., Scholes, M., 1973. The pricing of options and corporate liabilities. J. Polit. Econ. 81 (3), 637–654. https://doi.org/10.1086/260062.

Grosen, A., Jørgensen, P.L., 2002. Life insurance liabilities at market values: An analysis of insolvency risk, bonus policy, and regulatory intervention rules in a barrier option framework. J. Risk Insur. 69 (1), 63–91. https://doi.org/10.1111/1539-6975.00005.

Merton, R.C., 1973. Theory of rational option pricing. Bell J. Econ. Manage. Sci. 4, 141–183. https://doi.org/10.2307/3003143.

[1-8]. They assume in (vi) a capital-to-liability ratio to be 40% (p. 19) but no specific reason is given for this percentage. You should give a better explanation for this.

Response [1-8]:

1) One effort to efficiently reduce carbon emissions in production is through green technology investment.

(p. 20)

2) We apply Dermine and Lajeri (2001) to specify the high-emitter capital-to-liability ratio to be 40% since the low-emitter one is 30%, as mentioned above.

(p. 20)

[1-9]. You also consider in (vii) a bond interest rate of 2.41%. Is this an annual rate?

Response [1-9]:

Yes, this is an annual rate.

The yearly interest bond rate is 2.41%.

(p. 21)

[1-10]. Finally, the sample is poorly described. What is the period? When does it begin? And when does it end? How many observations were gathered? Also, the main summary statistics are missing as well as a normality test.”

Response [1-10]:

The response is the same as Response [1-3].

Our model is a one-period (i.e., one-year) one. The numerical analysis critically relies on the firm-level data collected. Toward that end, we apply the method used by Chen et al. (2022a), Chen et al. (2022b), Huang et al. (2022), and Li et al. (2023). Mainly, the firm-level parameters we collect for the numerical exercise are from different studies, and some are from assumptions based on theoretical reasonings.

(p. 19)

Reviewer #1

“This paper addresses the insurer green finance under regulatory cap-and-trade mechanism associated with green/polluting production during a war. This is a very interesting subject not yet addressed in literature, therefore deserving further attention. Congratulations! However, it should be improved in order to meet PLOS ONE standards. Thus, my comments to the authors are as follows:

[2-1]. Though the paper in general is written in understandable English some minor typos and grammatical errors need to be corrected.

Response [2-1]:

Thank you. We have double-checked the paper.

[2-2]. It is not clear why authors employ a contingent claim model. What are the advantages compared to alternative models?

Response [2-2]:

The contingent claims approach to the corporate securities valuation treats a firm’s equity as a call option on the firm’s assets (Black and Scholes, 1973) and a short put option on the firm’s value (Merton, 1973). Brockman and Turtle (2003) further formulate the corporate security valuation based on a path-dependent barrier option model, focusing on a premature structure. Briys and de Varenne (1994) use a contingent claim approach to evaluate a life insurance company’s equity. Grosen and Jørgensen (2002) apply Briys and de Varenne (1994) by introducing a premature state of an insurer’s equity valuation. This article develops the model in the spirit of Briys and de Varenne (1994), Brockman and Turtle (2003), and Grosen and Jørgensen (2002).

 (p. 5)

References

Black, F., Scholes, M., 1973. The pricing of options and corporate liabilities. J. Polit. Econ. 81 (3), 637–654. https://doi.org/10.1086/260062.

Grosen, A., Jørgensen, P.L., 2002. Life insurance liabilities at market values: An analysis of insolvency risk, bonus policy, and regulatory intervention rules in a barrier option framework. J. Risk Insur. 69 (1), 63–91. https://doi.org/10.1111/1539-6975.00005.

Merton, R.C., 1973. Theory of rational option pricing. Bell J. Econ. Manage. Sci. 4, 141–183. https://doi.org/10.2307/3003143.

[2-3]. They assume in (vi) a capital-to-liability ratio to be 40% (p. 19) but no specific reason is given for this percentage. Authors should give a better explanation for this.

Response [2-3]:

1) One effort to efficiently reduce carbon emissions in production is through green technology investment.

(p. 20)

2) We apply Dermine and Lajeri (2001) to specify the high-emitter capital-to-liability ratio to be 40% since the low-emitter one is 30%, as mentioned above.

(p. 20)

[2-4]. They also consider in (vii) a bond interest rate of 2.41%. Is this an annual rate?

Response [2-4]:

The yearly interest bond rate is 2.41%.

(p. 21)

[2-5]. Finally, the sample is poorly describel. What is the period? When does it begin? And when does it end? How many observations were gathered? Also, the main summary statistics are missing as well as a normality test.”

Response [2-5]:

Our model is a one-period (i.e., one-year) one. The numerical analysis critically relies on the firm-level data collected. Toward that end, we apply the method used by Chen et al. (2022a), Chen et al. (2022b), Huang et al. (2022), and Li et al. (2023). Mainly, the firm-level parameters we collect for the numerical exercise are from different studies, and some are from assumptions based on theoretical reasonings.

(p. 19)

Reviewer #2

“Dear Authors,

My pleasure to review your paper. I appreciate your work and have some comments and thoughts for your revision (if possible). 

In the introduction, the authors mentioned “this study aims to study the determinants of the life insurance policy’s optimal guaranteed rate setting and policyholder protection, considering credit risk from high-/low-carbon emission borrowing firm” which is the main aim of this study. However, in the data and findings, I was not able to understand/see appropriate firm-level data to detect our main research.

I appreciate the way you elaborate data and mathematical explanations, however it is quite lengthy in my personal opinion, while the table the authors provide data sources and selected variables seems not clear to me for my best understanding.

I suggest the following points hoping the authors could consider for revision:

The general structure is as follows:

[3-1]. Introduction- clearly state the context and main findings of the paper with main references (articles) the authors are contributing to the stream of related literature.

Response [3-1]:

Our four findings above demonstrate the influences of green loans, the regulatory cap of the cap-and-trade mechanism, affiliated reinsurance, and a war on the insurer’s performance in a borrowing-firm-insurer situation. Chen et al. (2022a) explore the impacts of the cap-and-trade transactions and financial gray rhino threats on insurer green performance. The research examines the effects of the government green subsidy with green trading on the insurer’s lending decisions (Chen et al., 2022b). Huang et al. (2022) analyze free riding and insurer carbon-linked investment. Li et al. (2023) investigate the green loan subsidy, regulatory cap, and green technology of borrowing-firm environmental impacts on insurer green finance assessment. Accordingly, our findings contribute to the literature on insurer green finance by extensively exploring the insurer’s green funding for the green and polluting borrowing firms participating in the cap-and-trade mechanism during a war.

(p. 6)

References

Chen, S., Huang, F.W., Lin, J.H., 2022a. Effects of cap-and-trade mechanism and financial gray rhino threats on insurer performance. Energies 15 (15), 5506. https://doi.org/10.3390/en15155506.

Chen, S., Huang, F.W., Lin, J.H., 2022b. Life insurance policyholder protection, government green subsidy, and cap-and-trade transactions in a black swan environment. Energy Econ. 115, 106333. https://doi.org/10.1016/j.eneco.2022.106333.

Huang, F.W., Chen, S., Lin, J.H., 2022. Free riding and insurer carbon-linked investment. Energy Econ. 17, 105838. https://doi.org/10.1016/j.eneco.2022.105838.

Li, X., Chen, L., Lin, J.H., 2023. Borrowing-firm environmental impact on insurer green finance assessment: Green loan subsidy, regulatory cap, and green technology. Environ. Impact Assess. Rev. 99, 107007. https://doi.org/10.1016/j.eiar.2022.107007.

[3-2]. Literature - could you please provide one seperate part to explain ‘the cap-and-trade’ mechanism for audiences to be more simpler in understanding this policy/mechanism.

Response [3-2]:

Carbon emission transactions in the cap-and-trade scheme are a cost-oriented mechanism for carbon emission reductions. The government determines a cap to incentivize firms to reduce carbon emissions. Firms are carbon-allowance buyers required by the government that their carbon allowance emissions exceed the regulatory cap. The carbon emission buying for production permission costs the high emitters, which creates an incentive to get involved in reducing carbon emissions in production. The carbon emission selling benefits the low emitters since they have put efforts into reducing carbon emissions. Carbon buyers and sellers in cap-and-trade transactions may improve environmental development by reducing carbon emissions.

(p. 7)

[3-3]. Data and methods - could you please clearly stay data vendor with main methods used for ths study and ‘why’s (could add some related papers to support your used methods). Please make the methodological.parts concise highlighting the main ones.

Response [3-3]:

Our model is a one-period (i.e., one-year) one. The numerical analysis critically relies on the firm-level data collected. Toward that end, we apply the method used by Chen et al. (2022a), Chen et al. (2022b), Huang et al. (2022), and Li et al. (2023). Mainly, the firm-level parameters we collect for the numerical exercise are from different studies, and some are from assumptions based on theoretical reasonings. It is well-recognized that other data selections yield different quantitative results, but the qualitative ones should remain the same.

(p. 19)

[3-4]. Findings - please highlight the main findings and explain how they contribute further to the literature (references you review in the introduction and literature review).

Response [3-4]:

Our four findings above demonstrate the influences of green loans, the regulatory cap of the cap-and-trade mechanism, affiliated reinsurance, and a war on the insurer’s performance in a borrowing-firm-insurer situation. Chen et al. (2022a) explore the impacts of the cap-and-trade transactions and financial gray rhino threats on insurer green performance. The research examines the effects of the government green subsidy with green trading on the insurer’s lending decisions (Chen et al., 2022b). Huang et al. (2022) analyze free riding and insurer carbon-linked investment. Li et al. (2023) investigate the green loan subsidy, regulatory cap, and green technology of borrowing-firm environmental impacts on insurer green finance assessment. Accordingly, our findings contribute to the literature on insurer green finance by extensively exploring the insurer’s green funding for the green and polluting borrowing firms participating in the cap-and-trade mechanism during a war.

 (p. 6)

[3-5]. Conclusion - could you please provide one paragraph for main context, one paragraph for main findings and one for future direction and policy.”

Response [3-5]:

We have developed a capped call option to explore the effects of insurer green lending, the regulatory cap of the cap-and-trade mechanism, affiliated reinsurance, and a war on the insurer’s performance. We show that increasing green loans toward borrowing-firm cleaner production decreases the insurer’s interest margin. The regulatory reduced cap raises the insurer’s margin, harming policyholder protection. The affiliated reinsurance reduces the insurer’s margin but benefits policyholder protection. Finally, increasing the war impact on the polluting borrowing firm makes the insurer more conservative by decreasing the insurance businesses. Our results should be attractive to investors, managers, and regulators.

As a market-oriented policy for controlling carbon emissions, the cap-and-trade mechanism brought new challenges to insurer green finance decisions, leading to cleaner production toward environmental sustainability. To face the challenges, we present two essential suggestions. We suggest the government green subsidy to stimulate the insurer to increase green lending toward environmental sustainability without hurting insurance stability. The government primarily focusing on decarbonization should also pay sufficient attention to ensuring climate and green finances, yielding financial stability. Therefore, our research complements the literature by providing two regulatory suggestions about sustainability practices, insurer operations, and insurance stability.

(pp. 28~29)

---

## [Decision Letter · Decision Letter 1]

27 Feb 2023

Insurer green finance under regulatory cap-and-trade mechanism associated with green/polluting production during a war

PONE-D-22-29807R1

Dear Dr. Lin,

We’re pleased to inform you that your manuscript has been judged scientifically suitable for publication and will be formally accepted for publication once it meets all outstanding technical requirements.

Kind regards,

Vu Quang Trinh, PhD

Academic Editor

PLOS ONE

Additional Editor Comments (optional):

Both reviewers and I agree that the paper has greatly improved and meets the journal standards. Therefore, I recommend an acceptance of the paper in its current form.

Reviewers' comments:

Reviewer's Responses to Questions

**Comments to the Author**

1. If the authors have adequately addressed your comments raised in a previous round of review and you feel that this manuscript is now acceptable for publication, you may indicate that here to bypass the “Comments to the Author” section, enter your conflict of interest statement in the “Confidential to Editor” section, and submit your "Accept" recommendation.

Reviewer #1: All comments have been addressed

Reviewer #2: All comments have been addressed

2. Is the manuscript technically sound, and do the data support the conclusions?

Reviewer #1: Yes

Reviewer #2: Yes

3. Has the statistical analysis been performed appropriately and rigorously? 

Reviewer #1: Yes

Reviewer #2: Yes

4. Have the authors made all data underlying the findings in their manuscript fully available?

Reviewer #1: No

Reviewer #2: Yes

5. Is the manuscript presented in an intelligible fashion and written in standard English?

Reviewer #1: Yes

Reviewer #2: Yes

6. Review Comments to the Author

Reviewer #1: After the revisions made by the authors I think the paper has greatly improved and meets the journal standards. Therefore, I suggest that it should be acepted.

Reviewer #2: (No Response)

7. PLOS authors have the option to publish the peer review history of their article (what does this mean?). If published, this will include your full peer review and any attached files.

Reviewer #1: No

Reviewer #2: No

---

## [Editor Report · Acceptance letter]

6 Mar 2023

PONE-D-22-29807R1 

Insurer green finance under regulatory cap-and-trade mechanism associated with green/polluting production during a war 

Dear Dr. Lin:

I'm pleased to inform you that your manuscript has been deemed suitable for publication in PLOS ONE. Congratulations! Your manuscript is now with our production department. 

Kind regards, 

on behalf of

Dr. Vu Quang Trinh 

Academic Editor

PLOS ONE